# Insights into *Haemophilus* macrolide resistance: A comprehensive systematic review and meta-analysis

Irfan Ahmad[1], Aziz Kubaev[2], Ahmed Hussein Zwamel[3,4,5], Roopashree R.[6], Lalji Baldaniya[7] Jaswinder kaur[8], Bindu Rani[9], Masoumeh Beig [10,11]*

1 Department of Clinical Laboratory Sciences, College of Applied Medical Sciences, King Khalid University, Abha, Saudi Arabia, 2 Department of Maxillofacial Surgery, Samarkand State Medical University, Uzbekistan, 3 Medical Laboratory Technique College, the Islamic University, Najaf, Iraq, 4 Medical Laboratory Technique College, the Islamic University of Al Diwaniyah, Al Diwaniyah, Iraq, 5 Medical Laboratory Technique College, the Islamic University of Babylon, Babylon, Iraq, 6 Department of Chemistry and Biochemistry, School of Sciences, JAIN, Bangalore, Karnataka, India, 7 Marwadi University Research Center, Department of Pharmacy, Faculty of Health Sciences, Marwadi University, Rajkot, Gujarat, India, 8 Department of Medical Lab Sciences, Chandigarh Group of Colleges-Jhanjeri, Punjab, India, 9 Department of Medicine, National Institute of Medical Sciences, NIMS University Rajasthan, Jaipur, India, 10 Department of Bacteriology, Pasteur Institute of Iran, Tehran, Iran, 11 Student Research Committee, Pasteur Institute of Iran, Tehran, Iran

* beigmasoumeh@gmail.com

## Abstract

### Background

*Haemophilus spp.*, particularly *Haemophilus influenzae*, are major global pathogens causing various infections. Macrolides are crucial in treating these infections, but rising resistance to macrolides in *Haemophilus spp.* highlights the growing threat of antimicrobial resistance (AMR).

### Objective

This study aims to assess the prevalence of macrolide resistance in *Haemophilus spp,* across different global regions.

### Methods

A systematic literature search was conducted across PubMed, Embase, Web of Science, and Scopus databases from May 2015 to December 2023 to identify studies on macrolide resistance in *Haemophilus* spp. The review included English-language full-text articles that reported resistance proportions and sample sizes. Study quality was assessed using the JBI Critical Appraisal Tool. Statistical analysis was performed using a random-effects model using the metafor package in R.

### Results

A total of 10,114 articles were retrieved, and after a comprehensive evaluation, 15 studies (from 19 reports) met the eligibility criteria for inclusion in this systematic review and

**Data availability statement:** All relevant data are in the manuscript and its supporting information files.

**Funding:** The author(s) received no specific funding for this work.

**Competing interests:** The authors have declared that no competing interests exist.

meta-analysis. Most studies (eight reports from three countries) focused on clarithromycin susceptibility, revealing a pooled prevalence of 7.2%. High heterogeneity was observed for azithromycin ($I^2$ = 96.31%, $p < 0.001$). Azithromycin resistance was higher than clarithromycin, with a resistance rate of 9.3% (nine reports), while erythromycin resistance was significantly higher at 79% (four reports). Subgroup analysis revealed significant variations in resistance prevalence based on geographic location and continent for azithromycin, erythromycin, and clarithromycin. Additionally, notable differences were observed in resistance rates depending on antimicrobial susceptibility testing (AST) methods and AST guidelines for both azithromycin and erythromycin. Clarithromycin resistance increased from 0.7% (2015–2019) to 12.6% (2020–2023).

## Conclusion

The study underscores the significant challenges of macrolide resistance in treating *Haemophilus* spp. infections. Additionally, ongoing surveillance of resistance patterns and exploring contributing factors are crucial to enhancing treatment effectiveness.

## Author summary

Antimicrobial resistance (AMR) threatens global public health, challenging the effectiveness of key antibiotics to treat infections. *Haemophilus* spp., particularly *Haemophilus influenzae*, are bacteria responsible for various serious illnesses, such as respiratory and ear infections. Macrolide antibiotics, including azithromycin, erythromycin, and clarithromycin, are essential to combating these infections. However, resistance to these drugs is increasing, making treatments less effective.

In this study, we analyzed data from published research to assess the prevalence of macrolide resistance in *Haemophilus* spp. globally. We found that resistance rates vary widely depending on the region, testing methods, and specific antibiotics studied. For example, azithromycin resistance was around 9.3%, while erythromycin resistance reached as high as 79%. Clarithromycin resistance has grown significantly in recent years, rising from 0.7% to 12.6% between 2020 and 2023.

These findings highlight the urgent need for ongoing monitoring of resistance patterns and improved strategies to manage infections caused by *Haemophilus* spp. By understanding how resistance evolves, healthcare providers can better tailor treatments, preserving the effectiveness of antibiotics and improving patient outcomes. This work underscores the broader challenge of AMR and the importance of global efforts to address this critical issue.

## 1.  Introduction

*Haemophilus influenzae* and *Haemophilus parainfluenzae* are essential members of the *Haemophilus* genus that contribute significantly to the respiratory tract and invasive infections [1]. *H. influenzae* is a notable pathogen that can lead to severe diseases, including pneumonia, meningitis, and otitis media [2]. Like *H. influenzae*, *H. parainfluenzae* also plays a critical role as a commensal organism of the upper respiratory tract and an opportunistic pathogen

in conditions such as endocarditis, bronchitis, and chronic obstructive pulmonary disease (COPD) exacerbations [3]. Including both species in this study is essential, as they share similar resistance mechanisms and clinical relevance. Macrolides, such as azithromycin and clarithromycin, are frequently used to treat respiratory tract infections caused by this organism [4]. The increasing prevalence of macrolide resistance in *Haemophilus spp.* especially *H. influenzae* and *H. parainfluenzae,* has raised significant concerns among healthcare professionals and public health officials [5].

Understanding the mechanisms and prevalence of this resistance is crucial for developing effective treatment strategies and mitigating the spread of resistant strains [6]. Resistance in these species is predominantly mediated by genetic factors, including the *erm* and *mefE* genes, which facilitate target site modification and drug efflux [7].

While azithromycin remains a key treatment option, comparing resistance trends is constrained by the limited data availability, particularly for *H. parainfluenzae*. Despite these constraints, existing evidence highlights worrying trends, including geographical variability in resistance rates and an overall increase in resistance prevalence. For instance, macrolide resistance rates for *H. influenzae* have been reported to reach as high as 30% in some regions, while *H. parainfluenzae* data remain sparse but suggest similar resistance mechanisms and trends.

The most common mechanism of macrolide resistance involves modification of the bacterial ribosomal target. This is predominantly mediated by the *erm* (erythromycin ribosome methylase) genes, which encode methyltransferases. These enzymes methylate adenine residues in the 23S ribosomal RNA of the 50S ribosomal subunit, reducing the binding affinity of macrolides. This leads to high-level resistance and cross-resistance to other macrolides, lincosamides, and streptogramin B (MLS_B phenotype). In *Haemophilus* spp., *erm* genes are often carried on mobile genetic elements, such as integrative conjugative elements or transposons, facilitating horizontal transfer [8]. The *mefE* gene is another significant contributor to macrolide resistance. It encodes an efflux pump that actively expels macrolide antibiotics from the bacterial cell, reducing intracellular concentrations and lowering the drug's efficacy. Resistance mediated by *mefE* is typically moderate and does not confer cross-resistance to other antibiotic classes. Efflux pump-mediated resistance is often plasmid-encoded, making it highly transmissible among bacterial populations [9].

Furthermore, mobile genetic elements such as plasmids and transposons facilitate the horizontal transfer of resistance genes, compounding the challenge of treating infections caused by these bacteria [10, 11]. Biofilm formation also plays a critical role in macrolide resistance, creating a physical barrier that limits antibiotic penetration and enhances the activity of efflux pumps [12]. Efflux pumps, such as AcrB in *H. influenzae*, contribute to multidrug resistance by actively expelling various antibiotics, complicating treatment strategies [11].

The shared resistance mechanisms between these species underscore the importance of addressing *H. influenzae* and *H. parainfluenzae* in surveillance and treatment strategies [13]. Further studies focused on *H. parainfluenzae* are needed to fill critical knowledge gaps, particularly in regions where data are sparse. Understanding the resistance patterns in these pathogens is crucial for guiding empirical antibiotic selection, optimizing treatment protocols, and informing antibiotic stewardship programs. These mechanisms not only complicate treatment regimens but also necessitate the need for ongoing surveillance of resistance patterns. Recent studies indicate a worrying trend in the prevalence of macrolide-resistant *H. influenzae* strains [5]. For instance, a systematic review reported that resistance rates vary significantly across different geographical regions, with some areas showing resistance rates as high as 30% [14].

In contrast, other regions, particularly in Asia, have reported lower resistance rates, suggesting that local antibiotic stewardship practices and the prevalence of resistant strains can differ markedly [15]. The rise in macrolide resistance necessitates reevaluating treatment

protocols for infections caused by *H. influenzae* [16]. Clinicians must consider local resistance patterns when selecting antibiotics, as empirical treatment based solely on in vitro susceptibility may lead to clinical failures [17]. Alternative antibiotics, such as respiratory fluoroquinolones or beta-lactams, may be warranted in cases where macrolide resistance is prevalent [18]. Furthermore, developing new antimicrobial agents and implementing combination therapies may provide additional avenues for effective treatment.

With dynamic changes in the epidemiology of *Haemophilus* spp. and the increasing issue of macrolide-resistant strains, a new approach is needed for effective management. Hence, the primary objective of this study was to systematically review and synthesize available data on the prevalence of macrolide resistance in *Haemophilus* spp. To ensure the data reflected current trends, we limited the search to studies published between 2015 and 2023.

The secondary objectives of this study were to identify trends and changes in resistance patterns over time and explore recent developments in *Haemophilus* spp. resistance, and investigate the impact of antibiotic usage and resistance patterns. Additionally, we aimed to explore heterogeneity in resistance rates across regions and populations and assess the influence of testing methods and guidelines on the reported resistance rates. By addressing these objectives, this study aims to fill existing knowledge gaps and provide comprehensive insights into the dynamics of macrolide resistance in *Haemophilus* spp., ultimately guiding future research and clinical practice.

Understanding the determinants of resistance development in these bacteria is pivotal for guiding antibiotic stewardship endeavors and optimizing treatment protocols for infections caused by *Haemophilus* spp. By rigorously analyzing available data, this study aimed to apprise healthcare practitioners and policymakers of the prevailing scenario of macrolide resistance in *Haemophilus* and identify potential strategies to address this burgeoning challenge.

## 2. Methods

Our study strictly followed the Preferred Reporting Items for Systematic Reviews and Meta-Analyses (PRISMA) guidelines [19] to ensure a comprehensive and rigorous meta-analytical synthesis of findings on macrolide resistance in *Haemophilus* spp. strains. Registration in the PROSPERO database (CRD42024565668) further demonstrates our commitment to methodological transparency and integrity.

### 2.1. Eligibility criteria

The meta-analysis's inclusion criteria required studies investigating macrolide resistance, specifically in Hemophilus species, with reported resistance proportions and providing sample size information. Only full-text articles published in English were considered eligible for inclusion. Exclusion criteria included studies written in languages other than English, case reports, clinical trials, and review articles. Studies with sample sizes of fewer than three isolates were excluded from the analysis. Additionally, antibiotics tested in fewer than three isolates were also excluded.

### 2.2. Information source and search strategy

A systematic search across the Scopus, PubMed, Web of Science, and EMBASE databases was conducted from 2015 to 2024, limited to articles published in English. The search syntax for each database was adapted according to the guidelines provided, utilizing the following terms: ("*Hemophilus*" OR "*H. influenzae*" OR "*H. parainfluenzae*" OR "*H. ducreyi*" OR "*H. haemolyticus*") AND ("Erythromycin" OR "Clarithromycin" OR "Azithromycin" OR "Dirithromycin" OR "Spiramycin" OR "Josamycin" OR "Roxithromycin" OR "Troleandomycin" OR "Clindamycin" OR "Telithromycin" OR "macrolide").

The search syntax for each database was adapted according to the guidelines provided. The search included relevant keywords such as macrolide, antibiotic resistance, *Hemophilus*, and related MeSH terms.

### 2.3. Selection process

Following a systematic database search, duplicate entries were removed using EndNote (version 21). To minimize bias, four authors (I-A, A-S-K, A-H-Z, and R-R, B-R) independently screened the titles, abstracts, and full texts of the identified publications to assess their eligibility for inclusion in the meta-analysis. Any discrepancies in the selection process were resolved by consultation with authors (L-B, J-K, and M-B), adjudicated, and the final decision was made.

### 2.4. Data collection process

Pertinent information, such as author(s), publication year, country, continent, diagnostic method, sample source, number of positive tests, and total sample size, was meticulously recorded during data extraction. Furthermore, specific data items in the analysis included information on antimicrobial susceptibility testing (AST) methods, adherence to relevant guidelines, year grouping, quality assessment grouping, *Haemophilus spp.*, and antibiotic resistance profiles of macrolides. Two independent authors (I-A and M-B) conducted the extraction to ensure data accuracy, resolving disparities through consensus.

### 2.5. Study risk of bias assessment

The quality of the included articles was assessed using the Joanna Briggs Institute (JBI) tool [20] by two independent authors (R-R, B-R), with discrepancies resolved by a third author (M-B). Each question in the tool was categorized into three levels of risk: low, some concern, and high. To determine the overall quality score, each risk level was assigned a numerical value: low risk [1], some concern (0.5), and high risk (0), with a maximum possible score of 10. Based on the total score, studies were classified into three categories: scores greater than 7 indicated high quality, scores between 4 and 6 signified some concern, and scores less than 4 indicated low quality. This scoring system provided a standardized approach for categorizing study quality and clarified distinguishing between high-quality, some-concern, and low-quality studies.

The quality of the included articles was assessed using the Joanna Briggs Institute (JBI) tool [20] by two independent authors (R-R, B-R), with discrepancies resolved by a third author (M-B). Each question in the tool was categorized into three levels of risk: low, some concern, and high. To determine the overall quality score, each risk level was assigned a numerical value: low risk (1), some concern (0.5), and high risk (0). An overall score greater than 7 indicates a high quality. Scores between 4 and 6 signified some concern, whereas scores less than 4 indicated low quality.

### 2.6. Effect measures

This meta-analysis evaluated the prevalence of antibiotic resistance by analyzing the proportion of resistant isolates reported across various studies. Subgroup analyses and meta-regression were conducted to identify factors contributing to variations in resistance rates, including geographic origin. Additionally, the study assessed temporal trends in macrolide resistance.

### 2.7. Synthesis methods

A random-effects model was applied, with the prevalence of antibiotic resistance as the primary outcome. Subgroup analysis and regression were conducted to explore sources of

heterogeneity, focusing primarily on differences by country and trends across years. A study was defined as a single article for this analysis, while a report referred to individual datasets or analyses within a study. A study could include one or more reports, each corresponding to distinct subpopulations, antibiotics, periods, or methodologies described within the article. To ensure consistency, data from all reports within a study were aggregated for study-level analyses while preserving the independence of findings across studies. This approach allowed for comprehensive analysis while minimizing bias introduced by multiple reports within a single study.

### 2.8. Statistics

Heterogeneity was assessed using the DerSimonian-Laird estimator, Q-test, and $I^2$ statistic. Meta-regression was used to analyze trends over time, while outlier detection employed studentized residuals and Cook's distance. The double arcsine transformation method was applied to stabilize variance and normalize proportions for meta-analysis. Funnel plot asymmetry was assessed using the rank correlation and regression tests. The analysis used R (version 4.2.1) and the metafor package (version 3.8.1) [21–28].

## 3. Result

### 3.1. Study selection

A total of 10114 records from the systematic search were imported into the reference manager software (EndNote version 21), from which 4583 duplicated articles were identified and subsequently removed. A total of 5531 articles were screened based on their title and abstract, resulting in 257 full-text articles being assessed. Following the evaluation, articles were excluded based on the predefined criteria. Ultimately, this systematic review and meta-analysis included 15 eligible studies [29–43] (details of the study characteristics and extracted data are available in S1 Table). The screening and selection processes are summarized in the PRISMA flowchart (Fig 1 and S2 Table). These studies originated from eight countries (the United States, China, Canada, Italy, Spain, Taiwan, Iran, and Japan), spanning three continents (North America, Asia, and Europe). The included studies were conducted between 2015 and 2023.

### 3.2. Study characteristics

Overall, the meta-analysis included a total of 15 studies conducted between the years 2015 and 2023 from various regions across the globe. Most of the reports were sourced from Asia ($n$ = 5) and America ($n$ = 5), followed by Europe ($n$ = 2). The studies included in this review were performed in eight countries: China, Japan, Iran, the United States, Italy, Spain, Canada, and Taiwan. Among the guidelines used to interpret antimicrobial susceptibilities, the Clinical and Laboratory Standards Institute (CLSI), European Committee on Antimicrobial Susceptibility Testing (EUCAST), and The British Society for Antimicrobial Chemotherapy (BSAC) were widely used in most studies (twelve, three, and one reports, respectively). The JBI critical appraisal checklist was utilized to evaluate the characteristics of the reviewed studies. Of the 15 included studies, 13 were low risk, and 2 were some risks.

### 3.3. Comprehensive overview of antibiotic resistance prevalence

The proportion of erythromycin resistance across the four reports, with 99 resistant isolates among 133 investigated isolates, was 0.79 (95% CI, 0.509, 0.932), exhibiting significant heterogeneity between reports ($I^2$ = 85.84%, p = 0.001). The proportion of clarithromycin resistance

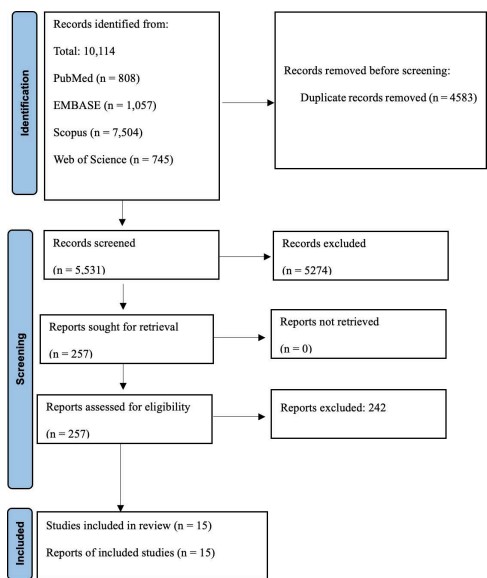

**Fig 1. The PRISMA flow chart provides an overview of the article selection process, detailing the steps of identification, screening, assessment for eligibility, and final inclusion of studies.** It highlights the number of records excluded at each stage and the reasons for exclusion, ensuring transparency and adherence to PRISMA guidelines.

across seven reports, with 219 resistant isolates among the 2700 investigated isolates, was 0.072 (95% CI, 0.029, 0.169), demonstrating significant heterogeneity between reports ($I^2$ = 95.86%, p= 0.001). Lastly, the proportion of azithromycin resistance across nine reports, with 4018 resistant isolates among the 13069 investigated isolates, was 0.093 (95% CI, 0.056, 0.15), which also showed significant heterogeneity between reports ($I^2$ = 96.31%, p = 0.001) (Fig 2).

**3.3.1. Prevalence of erythromycin resistance.** A total of 133 isolates derived from four studies were included in the analysis of erythromycin resistance. Using a random-effects model, the estimated average proportion was 0.790 (95% CI, 0.509–0.932), signifying a significant difference from zero (z = 2.013, p = 0.044). Heterogeneity between the studies was evident (Q(3) = 21.187, $I^2$ = 85.84%, p < 0.001). Application of the fill and trim method resulted in a proportion of 0.711 (95% CI, 0.431, 0.889). None of the studies were identified as outliers or overly influential, based on studentized residuals or Cook's distances. While the regression test indicated funnel plot asymmetry (p<0.001), the rank correlation test did not (p = 0.056) (Fig 3A).

**3.3.2. Prevalence of clarithromycin resistance.** A total of 2700 isolates investigated across eight studies were included in the analysis of clarithromycin resistance. Using a random-effects model, the estimated average proportion was 0.072 (95% CI, 0.029–0.169), demonstrating a significant difference from zero (z=-5.201, p<0.001). Substantial heterogeneity was observed between studies (Q(7) = 184.689, $I^2$=96.21%, p<0.001). After applying the fill and trim method, the proportion changed to 0.141 (95% CI, 0.059, 0.298). One study was identified as a potential outlier based on studentized residuals; however, after its exclusion, the proportion remained consistent. Cook's distances did not indicate any overly influential studies. Although the regression test showed funnel plot asymmetry (p=0.006), the rank correlation test did not (p=0.773) (Fig 3B).

**3.3.3. Prevalence of azithromycin resistance.** A total of 13069 isolates, investigated across ten studies, were included in analyzing azithromycin resistance. Using a random-effects model, the estimated average proportion was 0.093 (95% CI, 0.056–0.150), significantly differing from zero (z =- 8.203, p < 0.001). Significant heterogeneity between studies was

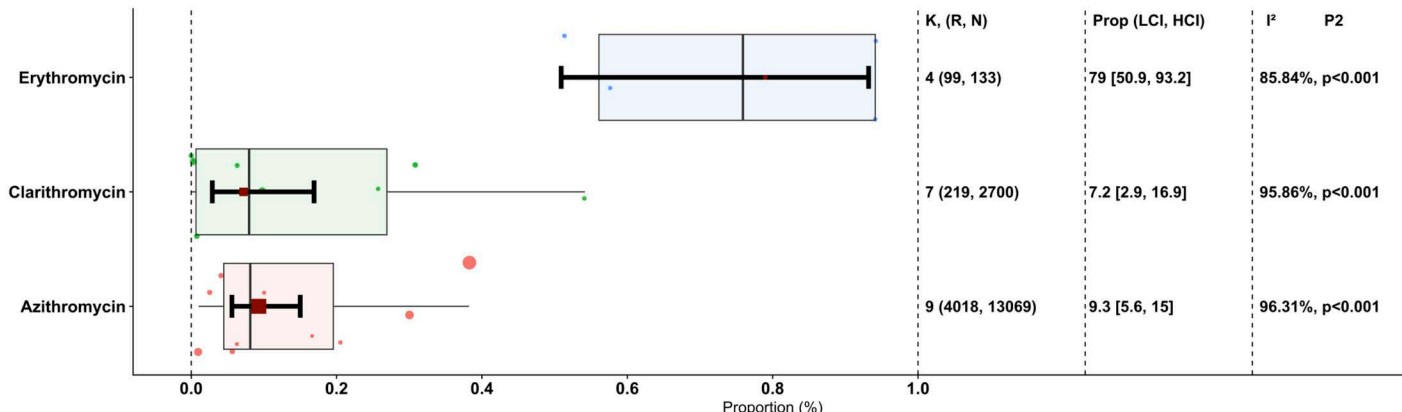

This visualization encapsulates the variance in antibiotic resistance rates, with each dot symbolizing an individual study's findings.
The dot size correlates with the study's sample size, providing insight into the data's robustness. Red square highlight the summarized resistance rates across studies, with error bars denoting confidence intervals.

**Fig 2. Forest plot depicting the overall proportion of *Hemophilus* isolates resistant to antibiotics (erythromycin, clarithromycin, azithromycin), calculated using a random-effects model.** Each antibiotic's resistance proportion is represented by a box plot, with error bars indicating 95% confidence intervals. The individual study results are shown as red points within each category, while the diamond shape at the bottom of each section represents the overall pooled estimate. The plot also includes heterogeneity statistics (I² and p-values), providing insights into variability across the included studies. This visualization highlights the variability in resistance rates across studies while providing an aggregated estimate for each antibiotic.

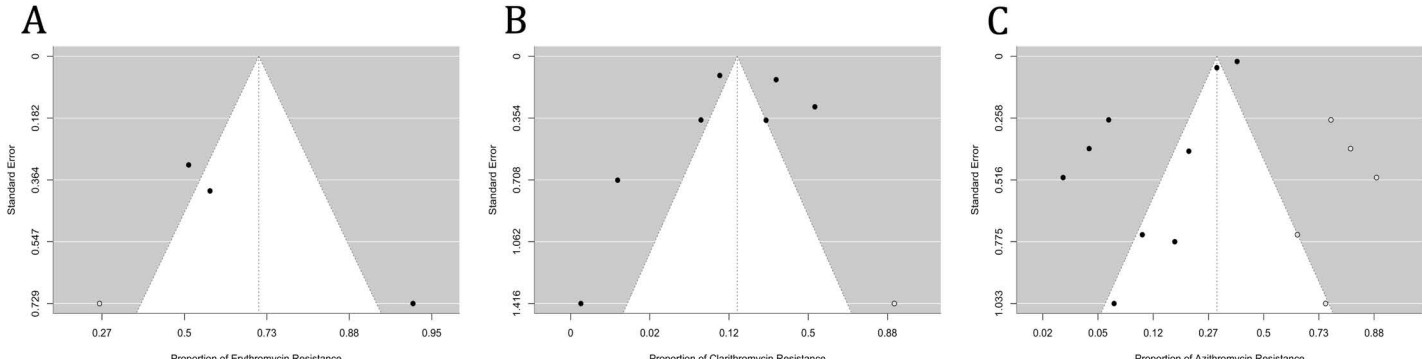

**Fig 3. Funnel plots assessing publication bias for resistance proportions of (A) erythromycin, (B) clarithromycin, and (C) azithromycin.** Each plot visualizes the relationship between the standard error (y-axis) and the proportion of resistance (x-axis) for individual studies included in the meta-analysis. The white region in each plot's center represents the symmetry area expected in the absence of publication bias, while the shaded areas indicate potential asymmetry. Symmetrical distributions suggest a low likelihood of publication bias, whereas asymmetry may indicate potential bias or heterogeneity in the included studies. These plots visually assess the reliability and robustness of the pooled estimates.

identified (Q(9) = 469.411, I² = 98.08%, p < 0.001). After applying the fill and trim method, the proportion remained unchanged at 0.093 (95% CI, 0.056, 0.150). One study was flagged as a potential outlier based on studentized residuals; however, after its exclusion, the proportion remained consistent. Cook's distances did not indicate any overly influential studies. While the regression test showed funnel plot asymmetry (p<0.001), the rank correlation test did not (p = 0.477) (Fig 3C).

### 3.4. Subgroup analysis

Subgroup analysis revealed significant variations in antibiotic resistance prevalence across countries (Fig 4). The analysis explored differences in resistance rates based on geographic regions, AST methods, temporal trends, and study quality (S3 Table).

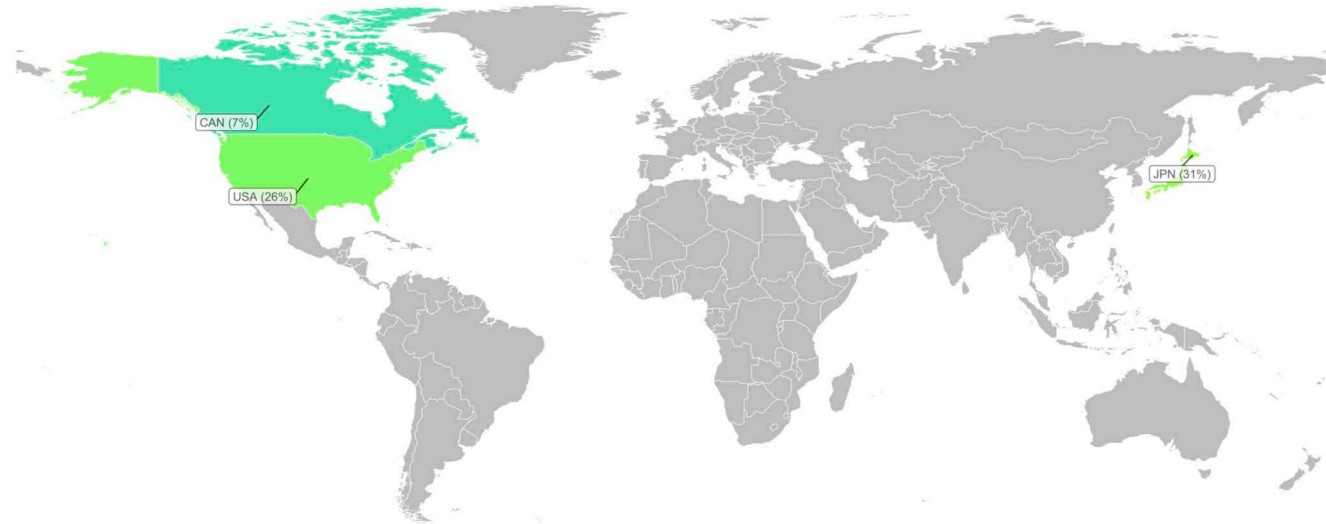

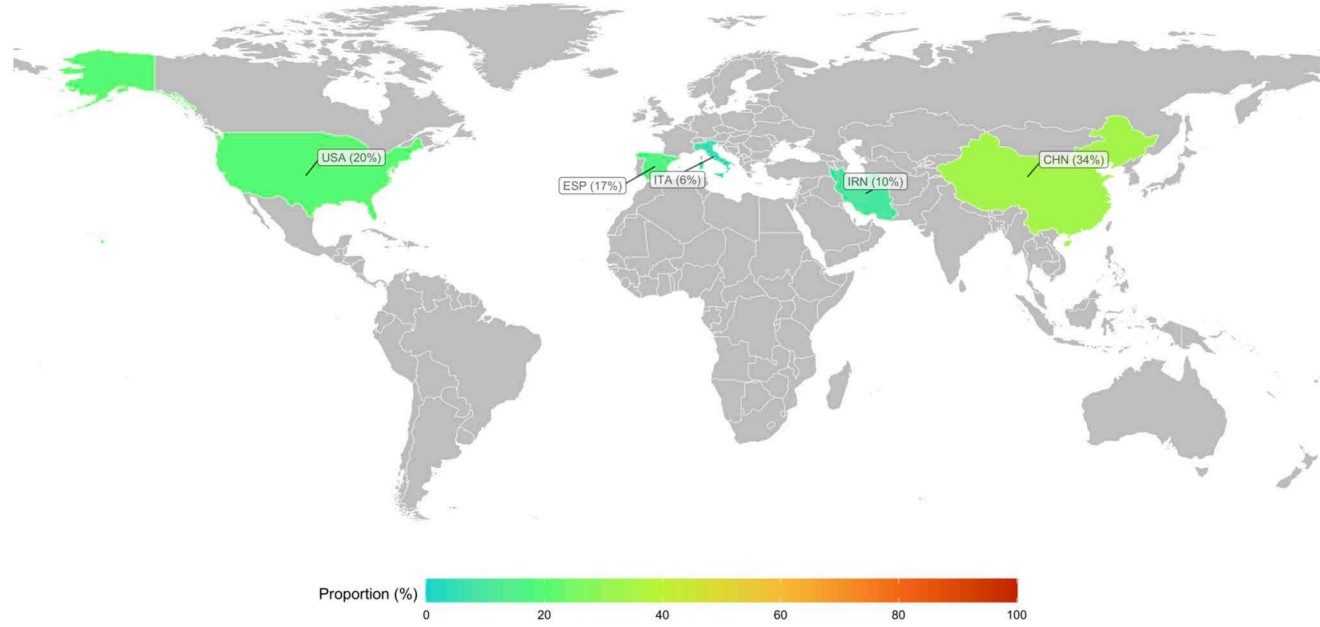

**Fig 4. Global prevalence of macrolide-resistant *Haemophilus* spp., illustrating resistance levels to (A) clarithromycin and (B) azithromycin across different regions.** The maps display the proportional resistance rates reported in each country, with colors ranging from green (low resistance) to orange (high resistance) to indicate the severity of resistance. Geographic disparities in resistance prevalence are evident, with certain regions, such as Asia, exhibiting higher resistance levels compared to North America and Europe. These maps provide a visual summary of the global distribution of macrolide resistance and underscore the need for region-specific antimicrobial stewardship and resistance surveillance efforts. Global map visualization was created using OpenStreetMap data, available under the Open Database License (ODbL).

**3.4.1. Subgroup analysis based on geographic regions.** Subgroup analysis revealed significant disparities in antibiotic resistance prevalence across different countries (Fig 4), particularly evident in azithromycin resistance rates. Italy exhibited the lowest resistance rate at 5.7%, while China reported the highest rate at 34%. Significant discrepancies in the prevalence of antibiotic resistance were also observed among continents, including azithromycin, clarithromycin, and erythromycin resistance rates (Fig 5A).

America (North America) displayed the lowest azithromycin resistance rate (2.4%), in contrast to Asia's highest rate of 32.2%. Similarly, North America exhibited the lowest clarithromycin resistance rate (0.7%), while Asia reported the highest rate at 30.8%. The Americas displayed the lowest resistance rate for erythromycin at 51.3%, whereas North America exhibited the highest rate at 94.1%. These variations in resistance rates can be attributed to regional differences in healthcare practices, antibiotic prescribing patterns, and antimicrobial stewardship programs. For instance, countries with stricter antibiotic usage policies and robust surveillance systems tend to report lower resistance rates. Conversely, regions with high antibiotic consumption, over-the-counter availability of antibiotics, or insufficient regulation of antibiotic use may experience higher resistance rates. The observed discrepancies highlight the importance of tailored strategies to combat antimicrobial resistance, considering local healthcare infrastructure, public health policies, and antibiotic consumption trends. These findings underscore the need for global collaboration to harmonize resistance surveillance efforts and promote judicious antibiotic use.

**3.4.2. Subgroup analysis based on antimicrobial susceptibility testing methods and guidelines.** Subgroup analysis revealed significant variations in antibiotic resistance prevalence across different AST methods, particularly in azithromycin and erythromycin resistance rates (Fig 5B). Notably, the multiple-method AST exhibited the lowest azithromycin resistance rate at 2% in contrast with the highest rate of 30.4% observed in the Disk Diffusion AST method. Similarly, the Disk Diffusion method displayed the lowest resistance rate for erythromycin at 51.3%, whereas no data were available for the highest rate.

Significant variations in the prevalence of antibiotic resistance were noted across different AST guidelines, particularly in azithromycin and erythromycin resistance rates (Fig 5C). The Multiple Guideline AST exhibited the lowest azithromycin resistance rate at 2%, in contrast to the highest rate of 31.2% observed in the CLSI guidelines. Similarly, the BSAC guidelines displayed the lowest resistance rate for erythromycin at 51.3%, while no data were available for the highest rate.

**3.4.3. Subgroup analysis across _Haemophilus spp._** Subgroup analysis based on species for all antibiotics investigated did not reveal statistically significant differences in the prevalence of macrolide resistance among _Haemophilus spp._ (Fig 5D). Comparisons between _H. influenzae_ and _H. parainfluenzae_ yielded p-values of 0.785 for erythromycin, 0.395 for clarithromycin, and 0.785 for azithromycin, indicating no significant variation in resistance rates between these species across the antibiotics studied.

**3.4.4. _Haemophilus_ macrolides resistance rate trend.** Furthermore, the analysis of resistance rates across different year groups (2015-2019 and 2020-2023) also did not show significant differences in the prevalence of macrolide resistance (Fig 5E). For erythromycin, clarithromycin and azithromycin, the p-values were 0.314, 0.080, and 0.x218, respectively. These findings suggest that the prevalence of macrolide resistance among _Haemophilus spp._ remains consistent, emphasizing the importance of continuously monitoring and implementing appropriate antibiotic stewardship measures to address this ongoing public health challenge.

## 3.5. Meta-regression

Meta-regression analysis explored the correlation between antibiotic resistance rates and the year of the report. For erythromycin, the correlation was not statistically significant (r =

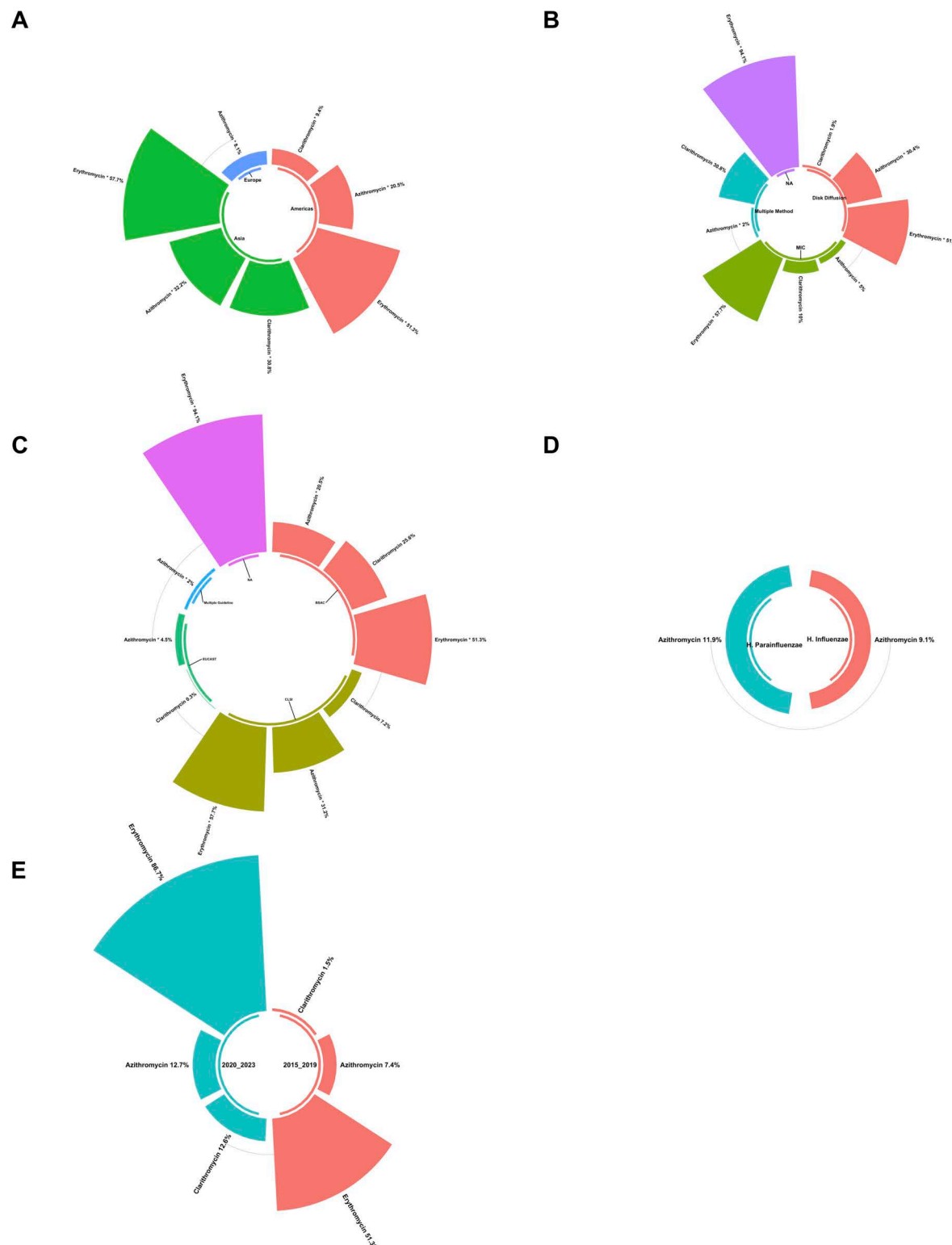

**Fig 5. Subgroup analysis results of macrolide-resistant *Haemophilus* spp., broken down by various factors.** (A) Comparison of prevalence by continent, illustrating regional differences in resistance levels. (B) Comparison of prevalence by AST methods, highlighting how different testing methodologies may influence resistance rates. (C) Comparison of prevalence by AST guidelines, showing how adherence to specific guidelines can affect reported resistance levels. (D) Comparison of prevalence among *Haemophilus* species, emphasizing

potential species-specific differences in resistance. **(E)** Comparison of prevalence before and after 2020, illustrating trends over time and the possible impact of recent public health interventions or shifts in antibiotic use. Each panel provides a detailed breakdown of the prevalence of resistance across these subgroups, offering insights into the factors driving the macrolide resistance in *Haemophilus* spp.

-0.087, p-value = 0.849, 95% CI [-0.975, 0.802]) (Fig 6A). This suggests that erythromycin resistance rates did not change significantly during the study period. In contrast, a significant positive trend was observed for clarithromycin (r = 0.639, p-value = 0.030, 95% CI [0.063, 1.215]) (Fig 6B). This indicates an increasing trend in clarithromycin resistance rates over time.

For azithromycin, the correlation was not statistically significant (r = 0.144, p-value = 0.464, 95% CI [-0.241, 0.529]) (Fig 6C). This suggests that azithromycin resistance rates remained relatively stable over the study period.

## 4. Discussion

*H. influenzae* and *H. parainfluenza* are clinically significant pathogens with shared resistance mechanisms, such as target site modification and efflux pumps [44]. Still, resistance trends for *H. parainfluenza* remain underexplored due to limited data availability, particularly for azithromycin [45]. Expanding surveillance efforts to include *H. parainfluenzae*, especially in underrepresented regions, is essential to understanding its resistance dynamics better and drawing meaningful comparisons with *H. influenzae*, guiding more effective treatment strategies and antibiotic stewardship programs [46].

The present systematic review and meta-analysis comprehensively assess macrolide resistance in *Haemophilus spp.*, elucidating the prevalence, geographic distribution, and methodological factors influencing resistance rates. These findings underscore the urgency of addressing this burgeoning public health concern and highlight the need for concerted efforts to mitigate the spread of resistant strains.

The meta-analysis revealed a substantial prevalence of macrolide resistance among *Haemophilus* isolates, with considerable heterogeneity between the studies. The estimated average proportions of resistance to erythromycin (79%), clarithromycin (7.2%), and azithromycin (9.3%) indicate the pervasiveness of this phenomenon. These findings corroborate the escalating macrolide resistance in *Haemophilus spp., as* reported in previous studies [47].

The observed heterogeneity in resistance rates can be attributed to several factors, including variations in study settings, geographical regions, AST methods, and adherence to guidelines. The subgroup analyses shed light on the influence of these factors on the prevalence of macrolide resistance.

Subgroup analysis based on geographic region revealed striking disparities in macrolide resistance rates across countries and continents. China had the highest azithromycin resistance rate (34%), whereas Italy had the lowest rate (5.7%). Similarly, significant differences were noted across continents, with Asia displaying the highest resistance rates to azithromycin (32.2%) and clarithromycin (30.8%), in contrast to North America's lowest rates (2.4% and 0.7%, respectively).

These geographical variations in macrolide resistance can be attributed to several factors, including antibiotic usage patterns, infection control practices, and dissemination of resistant strains within and across regions. Countries with higher antibiotic consumption rates and suboptimal infection control measures may facilitate the emergence and spread of resistant *Haemophilus* spp. Furthermore, the globalization of travel and trade can contribute to the international dissemination of resistant pathogens. For example, resistant strains have been reported to spread

Meta-regression Analysis: Proportion of **Erythromycin** Resistance Trends Over Time

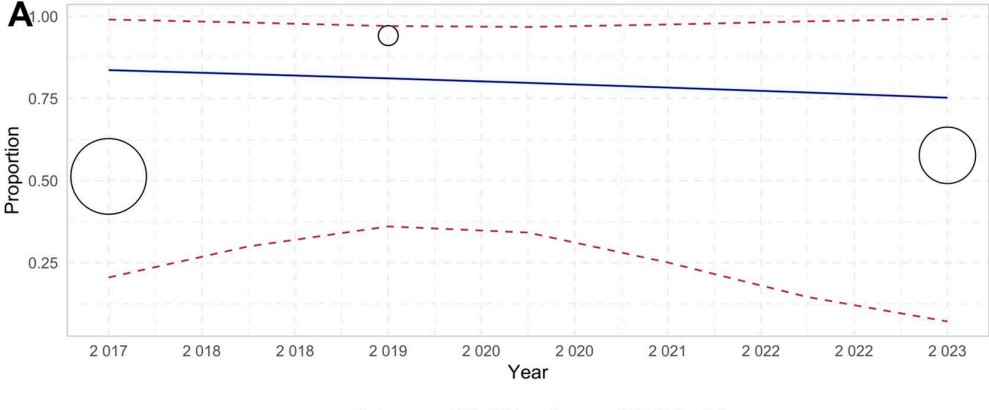

The correlation is not statistically significant (r = -0.087, *p*-value = 0.849, 95% CI [-0.975, 0.802]).

Meta-regression Analysis: Proportion of **Clarithromycin** Resistance Trends Over Time

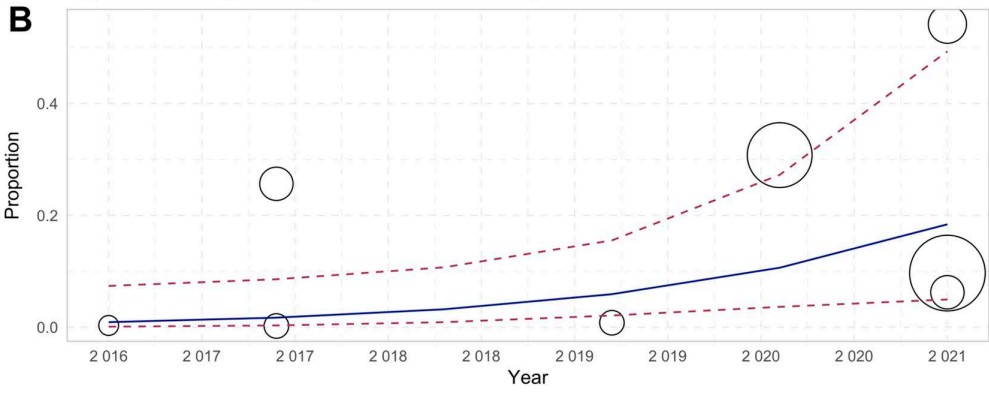

The correlation is statistically significant (r = 0.639, *p*-value = 0.030, 95% CI [0.063, 1.215]).

Meta-regression Analysis: Proportion of **Azithromycin** Resistance Trends Over Time

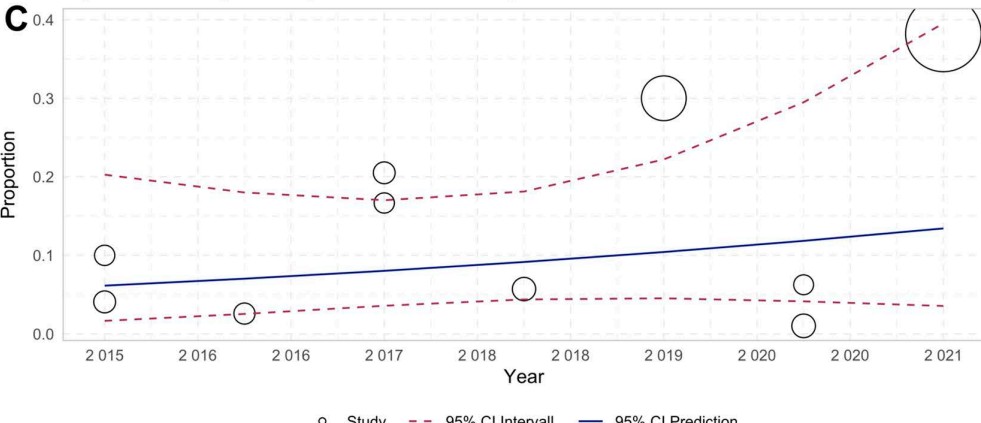

The correlation is not statistically significant (r = 0.144, *p*-value = 0.464, 95% CI [-0.241, 0.529]).

**Fig 6. Meta-regression analysis of macrolide resistance in *Haemophilus* spp. from 2015 to 2023. (A)** A scatter plot shows the trend of erythromycin-resistant isolates, demonstrating a stable resistance rate over time with a correlation coefficient of −0.087 and a non-significant p-value of 0.849. **(B)** A scatter plot depicts a significant upward trend in clarithromycin resistance, with a positive correlation coefficient of 0.639 and a statistically significant p-value of 0.03, indicating an increase in resistance over the years. **(C)** A scatter plot shows the trend of azithromycin-resistant isolates, with a stable resistance rate over time reflected by a correlation coefficient of 0.144 and a non-significant p-value of 0.464. Each data point represents a study, with the size of the circle indicating the study's weight in the

analysis. Solid blue lines represent fitted regression lines, and dashed red lines represent the 95% confidence intervals. These plots provide insights into temporal changes in macrolide resistance, highlighting differences in trends across the three antibiotics.

across continents through international travel, where individuals carrying resistant bacteria unknowingly transmit them to new regions [48]. Similarly, the global trade of agricultural products, such as livestock treated with antibiotics, can introduce resistant bacteria into countries with previously low resistance rates. These pathways highlight the critical need for coordinated international efforts to monitor and control the spread of antimicrobial resistance [49].

Subgroup analysis revealed significant disparities in the prevalence of macrolide resistance based on the AST method. The multiple AST method exhibited the lowest azithromycin resistance rate (2%), while the Disk Diffusion method displayed the highest rate (30.4%). These discrepancies highlight the importance of standardized and harmonized AST procedures for accurate resistance detection and surveillance.

Different AST methods may exhibit varying sensitivities and specificities in detecting resistant strains, leading to variations in the resistance rates [50]. The Disk Diffusion method is widely used because of its simplicity and cost-effectiveness. However, factors such as the inoculum density, incubation conditions, and interpretation criteria [51] may influence it. Conversely, molecular methods, such as those employed in the Multi-Method AST, offer higher specificity and can detect resistance mechanisms at the genetic level [52].

Subgroup analysis based on adherence to the AST guidelines revealed notable differences in the prevalence of macrolide resistance. The Multiple Guideline AST exhibited the lowest azithromycin resistance rate (2%), whereas the CLSI guideline displayed the highest rate (31.2%). These findings underscore the importance of adhering to standardized guidelines for accurate and consistent resistance detection.

Differences in interpretative criteria, quality control procedures, and recommended testing methodologies [53] may cause variations in AST guidelines. Adherence to internationally recognized guidelines, such as those provided by the CLSI or the EUCAST, can promote harmonization and facilitate accurate resistance surveillance [54].

Standardizing AST methods is essential to reducing variability and ensuring the comparability of resistance data across laboratories and studies. Variability in AST methods, such as Disk Diffusion, molecular techniques, or broth microdilution, can lead to inconsistent detection of resistance due to differences in sensitivity, specificity, and interpretative criteria [55]. Similarly, variations in guidelines, such as those provided by CLSI and EUCAST, may result in differing resistance thresholds, complicating comparisons across regions and studies. These discrepancies have significant implications for both clinical practice and policymaking. In clinical settings, resistance detection inconsistencies can affect susceptibility reporting accuracy, potentially leading to inappropriate antibiotic selection and suboptimal patient outcomes. For policymakers, discrepancies in resistance surveillance data may misrepresent actual resistance trends, hindering the development of effective public health interventions and antimicrobial stewardship programs.

Harmonizing AST methods and standardizing interpretative guidelines are critical to address these challenges. This can be achieved by adopting internationally recognized guidelines such as CLSI or EUCAST, harmonizing breakpoints across these guidelines, and enforcing strict quality control measures, including using standardized media, reagents, and control strains [56]. Incorporating molecular testing methods, such as PCR and whole-genome sequencing, alongside phenotypic tests can enhance the detection of resistance mechanisms. Standardized reporting formats with transparent documentation of procedures and criteria

are essential for consistent data interpretation. Global collaboration through centralized databases and regular training for laboratory personnel can further enhance data reliability and promote the sharing of resistance trends. Developing simplified yet reliable tools for low-resource settings is necessary to ensure equitable participation in global resistance surveillance efforts. These measures collectively contribute to accurate resistance monitoring and effective policymaking in the fight against antimicrobial resistance.

A meta-regression analysis examined the relationship between resistance rates and publication years, identifying a statistically significant correlation for one of the three antibiotics tested—clarithromycin. This finding highlights a potential increase in macrolide resistance in *Haemophilus* spp., consistent with the global trend of rising antimicrobial resistance [57]. However, no significant correlations were observed for the other two macrolides analyzed.

The escalation of macrolide resistance can be attributed to several factors, including inappropriate antibiotic prescription practices, suboptimal infection control measures, and selective pressure exerted by antibiotic use [58]. Additionally, the ability of *Haemophilus* spp. to acquire and disseminate resistance mechanisms through horizontal gene transfer and clonal expansion can contribute to the rapid spread of resistant strains [59]. The findings of this systematic review and meta-analysis have significant implications for clinical practice, public health policies, and future research. The prevalence of macrolide resistance in *Haemophilus spp.* underscores the need for informed antibiotic stewardship programs and enhanced surveillance efforts to monitor resistance trends [60].

For public health policies, specific recommendations include establishing standardized guidelines for AST and reporting to ensure consistent resistance data across regions [61]. Policymakers should invest in strengthening surveillance systems, particularly in low-resource settings, to capture comprehensive resistance patterns [62]. Additionally, stricter regulations on over-the-counter antibiotic sales and campaigns to promote the rational use of antibiotics in both healthcare and agriculture are crucial [49]. International collaboration is also needed to track the global spread of resistant strains and harmonize policies for antimicrobial use. These measures can support the development of targeted interventions to mitigate the growing threat of antimicrobial resistance [63].

Healthcare professionals should exercise caution when prescribing macrolides to treat *Haemophilus* infections, particularly in regions with high resistance rates. Based on local resistance patterns and clinical guidelines, alternative therapeutic options, such as third-generation cephalosporins or fluoroquinolones, may be considered [64].

Moreover, the observed geographical variations in macrolide resistance highlight the importance of tailored interventions and regional strategies to combat resistance. Countries and regions with higher resistance rates may require more stringent infection control measures, antibiotic stewardship programs, and public awareness campaigns to curb the spread of resistant strains [65].

Standardization of AST methods and adherence to recognized guidelines are crucial for accurate resistance detection and surveillance. Harmonizing AST procedures and interpretative criteria can facilitate global monitoring efforts and inform treatment and policy development [66] decision-making.

Future research should focus on elucidating the underlying mechanisms driving macrolide resistance in *Haemophilus spp.*, including identifying resistance genes, mobile genetic elements, and potential reservoirs of resistant strains. Additionally, to mitigate the consequences of this growing public health threat, it is warranted to investigate the impact of macrolide resistance on clinical outcomes and explore alternative therapeutic strategies.

This meta-analysis has certain limitations that must be acknowledged. One major limitation is the reliance on published data, which may not fully capture the true resistance rates

in all regions. For example, resistance to a particular macrolide might exist in a region but remain undocumented if no studies have been conducted or published. This introduces the possibility of geographic or reporting bias, potentially affecting the comprehensiveness of our findings. Additionally, variations in study design, sample sizes, and methodologies among the included studies may lead to heterogeneity in the data, influencing the generalizability of the results. Furthermore, the temporal scope of the studies may not fully reflect current resistance trends, as data from some regions may be outdated. These limitations highlight the need for global, more comprehensive, and standardized antimicrobial resistance surveillance.

Despite the comprehensive search strategy employed in this meta-analysis, including only 15 studies represents a notable limitation. This limited dataset may impact the generalizability of the findings, particularly for regions like Africa and South America, where data were sparse or absent. The underrepresentation of these regions highlights the need for more robust research efforts to address gaps in surveillance and reporting. Expanding data collection in underrepresented areas would provide a more globally comprehensive understanding of macrolide resistance, enabling the development of targeted and effective public health interventions. This limitation is a critical factor in interpreting the study's findings and their broader implications.

## 5. Conclusion

This systematic review and meta-analysis revealed a notably high prevalence of macrolide resistance in *Haemophilus spp.*, particularly to erythromycin. Resistance rates varied significantly across regions, with erythromycin resistance ranging from 51.3% in the Americas to 94.1% elsewhere, clarithromycin resistance spanning 7.0% in Canada to 30.8% in Japan, and azithromycin resistance between 2.4% and 32.2%, indicating an upward trend. The findings emphasize the critical need for region-specific treatment strategies and the implementation of robust global surveillance systems. Such programs are vital for monitoring resistance patterns, investigating contributing factors like antibiotic usage and strain prevalence, and optimizing treatment outcomes. Future research should focus on these dynamics to develop effective strategies.

## Supporting information

**S1 Table.** Characteristics and extracted data of studies included in the meta-analysis. (DOCX)

**S2 Table.** Detailed PRISMA Checklist for Systematic Review Reporting. (DOCX)

**S3 Table.** Prevalence of Antibiotic Resistance. (DOCX)

## Acknowledgment

The authors thank the Deanship of Research and Graduate Studies, King Khalid University, Abha, Saudi Arabia, for supporting this work through the Large Research Group Project.

## Author contributions

**Conceptualization:** Irfan Ahmad, Masoumeh Beig.

**Data curation:** Irfan Ahmad, Ahmed Hussein Zwamel, Bindu Rani.

**Formal analysis:** Irfan Ahmad.

**Investigation:** Aziz Kubaev, Ahmed Hussein Zwamel, Roopashree R, Masoumeh Beig.

**Methodology:** Masoumeh Beig.

**Project administration:** Masoumeh Beig.

**Validation:** Roopashree R, Jaswinder kaur, Bindu Rani.

**Visualization:** Aziz Kubaev, Ahmed Hussein Zwamel, Lalji Baldaniya, Jaswinder kaur.

**Writing – original draft:** Irfan Ahmad, Aziz Kubaev, Ahmed Hussein Zwamel, Roopashree R, Lalji Baldaniya, Jaswinder kaur, Masoumeh Beig.

**Writing – review & editing:** Irfan Ahmad, Masoumeh Beig.

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
