## [Decision Letter · Decision Letter 0]

6 Jan 2025

PNTD-D-24-01839Insights into Hemophilus Macrolide Resistance: A Comprehensive Systematic Review and Meta-AnalysisPLOS Neglected Tropical Diseases Dear Dr.Beig, Thank you for submitting your manuscript to PLOS Neglected Tropical Diseases. After careful consideration, we feel that it has merit but does not fully meet PLOS Neglected Tropical Diseases's publication criteria as it currently stands. Therefore, we invite you to submit a revised version of the manuscript that addresses the points raised during the review process. Please submit your revised manuscript within 30 days Mar 07 2025 11:59PM. If you will need more time than this to complete your revisions, please reply to this message or contact the journal office at plosntds@plos.org. Please include the following items when submitting your revised manuscript: * A rebuttal letter that responds to each point raised by the editor and reviewer(s). You should upload this letter as a separate file labeled 'Response to Reviewers '. This file does not need to include responses to any formatting updates and technical items listed in the 'Journal Requirements' section below. * A marked-up copy of your manuscript that highlights changes made to the original version. You should upload this as a separate file labeled 'Revised Manuscript with Track Changes '. * An unmarked version of your revised paper without tracked changes. You should upload this as a separate file labeled 'Manuscript '. If you would like to make changes to your financial disclosure, competing interests statement, or data availability statement, please make these updates within the submission form at the time of resubmission. Guidelines for resubmitting your figure files are available below the reviewer comments at the end of this letter. We look forward to receiving your revised manuscript.

Kind regards,

Prashant Kumar, Ph.D.Academic EditorPLOS Neglected Tropical Diseases

Ana LTO Nascimento

Section Editor

Shaden Kamhawi

co-Editor-in-Chief

Paul Brindley

co-Editor-in-Chief

**Journal Requirements:**

At this stage, the following Authors/Authors require contributions: Bindu Rani. Please ensure that the full contributions of each author are acknowledged in the "Add/Edit/Remove Authors" section of our submission form.

4) We have noticed that you have uploaded Supporting Information files, but you have not included a complete list of legends. Please add a full list of legends for your Supporting Information files after the references list.

Potential Copyright Issues:

- Figure 4; Please provide a direct link to the base layer of the map (i.e., the country or region border shape) and ensure this is also included in the figure legend; and provide a link to the terms of use / license information for the base layer image or shapefile. We cannot publish proprietary or copyrighted maps (e.g. Google Maps, Mapquest) and the terms of use for your map base layer must be compatible with our CC BY 4.0 license.

**Reviewers' comments:** Reviewer's Responses to Questions

**Key Review Criteria Required for Acceptance?**

**Methods**

-Are the objectives of the study clearly articulated with a clear testable hypothesis stated?

-Is the study design appropriate to address the stated objectives?

-Is the population clearly described and appropriate for the hypothesis being tested?

-Is the sample size sufficient to ensure adequate power to address the hypothesis being tested?

-Were correct statistical analysis used to support conclusions?

-Are there concerns about ethical or regulatory requirements being met?

Reviewer #1: (No Response)

Reviewer #2: (No Response)

Reviewer #3: (No Response)

Reviewer #4: Yes, to all except last bullet point. No ethical or regulatory concerns.

**Results**

-Does the analysis presented match the analysis plan?

-Are the results clearly and completely presented?

-Are the figures (Tables, Images) of sufficient quality for clarity?

Reviewer #1: (No Response)

Reviewer #2: (No Response)

Reviewer #3: (No Response)

Reviewer #4: Overall, yes. Image quality was not the best especially for Figure 5. I did not see an option to download high quality images. So, please ensure that the image quality is good prior to publishing the manuscript.

**Conclusions**

-Are the conclusions supported by the data presented?

-Are the limitations of analysis clearly described?

-Do the authors discuss how these data can be helpful to advance our understanding of the topic under study?

-Is public health relevance addressed?

Reviewer #1: (No Response)

Reviewer #2: (No Response)

Reviewer #3: (No Response)

Reviewer #4: Yes, to all.

**Editorial and Data Presentation Modifications?**

Reviewer #1: (No Response)

Reviewer #2: (No Response)

Reviewer #3: (No Response)

Reviewer #4: 1. Abstract, Line 51: “Clarithromycin resistance increased from q.5% (2015–2019) to 12.6% (2020–2023)”. Please check the number q.5%

2. Selection Process, Lines 129-130: “Following a systematic database search, duplicates were removed using EndNote (version 21), and duplicate entries were removed”. Rewrite sentence to remove redundancy.

3. Section 2.4 and 2.5: Information in both these sections seems redundant? Can this be merged?

4. Line 151-152: “S scores between 4 and 6 signified some concern, whereas scores less than 4 indicated low quality”. The first S seems like a typo. Please correct if that’s the case.

5. Section 3.1 Lines 176-127: “These studies originated from eight countries (the United States, China, Canada, Italy, Spain, Taiwan, Iran, and Japan) spanning four continents (North America, America, Asia, and Europe). The included studies will be conducted between 2015 and 2023” The countries listed span only three continents. America is not a continent. Please correct since North America is already included.

6. Section 3.2 Lines 184-187: “Among the guidelines used to interpret antimicrobial susceptibilities, CLSI (Clinical & Laboratory Standards Institute), EUCAST (European Committee on Antimicrobial Susceptibility Testing), and BSAC (The British Society for Antimicrobial Chemotherapy) were widely used in most studies (12, four, and one reports, respectively)”. Please be consistent in the usage of how the numbers are depicted i.e., 12 vs. four and one).

7. Figure 2: There is footnote embedded in the figure. Please include that in the legend of the Figure.

8. Section 3.4.1 Lines 237-243 and Figure 5A: Since the data for North America is being discussed it would be best if the America’s data (which I assume is North and South America combined) be split to be able to relate the text to Figure 5A.

9. Figure 5 was extremely hard to review because of the quality of the figure. Please ensure that the high-resolution figure gets published in the final manuscript. Usually, there is an option to download high-resolution figures during the review process, which I believe was not provided here.

10. Discussion Lines 328-331: “A meta-regression analysis explored the relationship between macrolide resistance rates and publication years and revealed a statistically significant correlation. This finding suggests that macrolide resistance in Haemophilus spp. May increase, aligning with the global trend of rising antimicrobial resistance (43).” I think it will be worth highlighting that a correlation was only observed for one (Clarithromycin) of the three antibiotics tested.

11. Please discuss the limitations of the type of meta-analysis performed in this study. For example, what if there is resistance to a particular macrolide in a region but that has never been studied. How do we account for that? Please add additional limitations as deemed fit.

**Summary and General Comments**

Reviewer #1: (No Response)

Reviewer #2: The study’s focus on clarithromycin, azithromycin, and erythromycin excludes newer or less commonly used macrolides, potentially limiting the comprehensive understanding of resistance dynamics. A more inclusive approach would involve investigating macrolides and, if applicable, other antibiotics from the same class. This should not be restricted to specific antibiotics.

Differences in resistance rates between AST methods and guidelines are noted but not adequately contextualized in terms of their implications for clinical practice and policymaking.

Although mechanisms such as erm and mefE genes are briefly mentioned, the discussion could benefit from a deeper exploration of genetic and molecular factors driving macrolide resistance.

Provide detailed recommendations for standardizing AST methods to reduce variability and enhance comparability of resistance data.

Line 313: Minor grammatical errors and awkward phrasing (e.g., “red resistance rates”) should be corrected for clarity.

Line 34: Replace “pooled random-effects model in R, utilizing the metafor package” with “random-effects model using the metafor package in R” for conciseness.

Line 52: The sentence “The high prevalence and regional variability of resistance highlight…” is slightly repetitive of earlier content. Revise to emphasize the findings’ implications.

Line 69-71: Replace “thereby reducing the binding affinity of macrolides” with “leading to decreased macrolide efficacy” for brevity.

Line 114-116: Rephrase “studies with sample sizes of less than three isolates and antibiotics with fewer than three isolates were excluded” for clarity. Consider splitting it into two sentences.

Line 236-243 (Subgroup Analysis by Region): Provide additional context for resistance variability, including differences in healthcare practices or antibiotic use policies.

Line 301-304: Include a specific example illustrating how “globalization of travel and trade” may contribute to the spread of resistance.

Line 328-329: Rephrase “a statistically significant correlation” as “a significant positive trend” to align with the tone of the remainder of the discussion.

Line 337-340: Expand on the “implications for public health policies” with specific recommendations for policy-makers.

References: Ensure consistent formatting of journal names and volume/issue numbers.

Tables: Supplementary tables should be cross-referenced in the main text for ease of access.

Figure 3: Labels have been added to clarify the axes for readers unfamiliar with meta-analysis visualizations.

Use more intuitive and comprehensive captions to figures.

Reviewer #3: This study evaluates macrolide resistance in Haemophilus species globally through a systematic review and meta-analysis. It identifies significant resistance trends, highlights regional variability, and emphasizes the need for localized treatment strategies and enhanced surveillance. Advanced statistical methods, including meta-regression and random-effect modeling, are employed to ensure reliable conclusions. The study’s adherence to PRISMA guidelines and PROSPERO registration ensures transparency and reproducibility, making it a valuable resource for addressing antimicrobial resistance challenges, especially in underrepresented regions.

There are several comments and suggestions for improvement outlined below to enhance the study's comprehensiveness and impact.

Comments for Improvement

General Comments

1. Study Scope and Limitations:

o Although a comprehensive search strategy was performed, the inclusion of only 15 studies is a notable limitation. This could affect the generalizability of findings, especially for underrepresented regions like Africa and South America where data were sparse or missing. Please acknowledge this limitation explicitly in the manuscript and discuss its implications.

o The focus on clarithromycin, azithromycin, and erythromycin excludes newer or less commonly used macrolides. Expanding the analysis to include additional macrolides or related antibiotics would provide a more comprehensive understanding of resistance dynamics.

2. AST Methods and Clinical Implications:

o Differences in resistance rates across antimicrobial susceptibility testing (AST) methods and guidelines are noted but not adequately contextualized. Discuss how these differences impact clinical decision-making and policymaking.

3. Species-Specific Discussion:

o The inclusion of Haemophilus influenzae and Haemophilus parainfluenzae is valuable, but these species are not sufficiently discussed in the introduction and discussion. Provide more context about their relevance and resistance trends, particularly since the resistance comparison for azithromycin is limited due to data constraints.

Methodological Clarifications

1. Clearly define and justify the study’s definition of terms and reporting criteria in the Methods section.

2. Provide additional details about statistical techniques used, such as back transformation, double arcsin transformation, and logistic regression, to enhance transparency.

3. For the Risk of Bias Assessment (Lines 146-152), clarify how "some concern" and "low quality" ratings were classified, and include thresholds for these classifications.

Minor Comments

1. Remove unnecessary lines, such as Lines 22-23.

2. Lines 33-34 are repeated from the Methods section; please remove them.

3. Add Haemophilus influenzae as a keyword to improve indexing.

4. Ensure consistent font color throughout the manuscript (use black for all text).

5. Line 187: Adjust numerical formatting to ensure consistency.

6. Line 291: Verify and correct reference formatting.

7. Line 324: Add appropriate spacing before and after Reference 41.

8. Lines 325-326: Use only abbreviations for the Clinical and Laboratory Standards Institute (CLSI) and the European Committee on Antimicrobial Susceptibility Testing (EUCAST) after first mention.

Reviewer #4: Overall, I enjoyed reading the publication. I have a few minor comments. Please refer to that section for the comments.

PLOS authors have the option to publish the peer review history of their article (what does this mean? ). If published, this will include your full peer review and any attached files.

**Do you want your identity to be public for this peer review?** For information about this choice, including consent withdrawal, please see our Privacy Policy .

Reviewer #1: **Yes: ** Safoura Moradkasani

Reviewer #2: No

Reviewer #3: No

Reviewer #4: No

**Figure resubmission:** While revising your submission, please upload your figure files to the Preflight Analysis and Conversion Engine (PACE) digital diagnostic tool, https://pacev2.apexcovantage.com/. PACE helps ensure that figures meet PLOS requirements. To use PACE, you must first register as a user. Registration is free. Then, login and navigate to the UPLOAD tab, where you will find detailed instructions on how to use the tool. If you encounter any issues or have any questions when using PACE, please email PLOS at figures@plos.org. Please note that Supporting Information files do not need this step. If there are other versions of figure files still present in your submission file inventory at resubmission, please replace them with the PACE-processed versions.**Reproducibility:** To enhance the reproducibility of your results, we recommend that authors of applicable studies deposit laboratory protocols in protocols.io, where a protocol can be assigned its own identifier (DOI) such that it can be cited independently in the future. Additionally, PLOS ONE offers an option to publish peer-reviewed clinical study protocols. Read more information on sharing protocols at https://plos.org/protocols?utm_medium=editorial-email&utm_source=authorletters&utm_campaign=protocols

---

## [Decision Letter · Decision Letter 1]

28 Jan 2025

Dear dr Beig,

We are pleased to inform you that your manuscript 'Insights into Haemophilus Macrolide Resistance: A Comprehensive Systematic Review and Meta-Analysis' has been provisionally accepted for publication in PLOS Neglected Tropical Diseases.

Best regards,

Prashant Kumar, Ph.D.

Academic Editor

Elsio Wunder Jr

Section Editor

Shaden Kamhawi

co-Editor-in-Chief

Paul Brindley

co-Editor-in-Chief

Reviewer's Responses to Questions

**Key Review Criteria Required for Acceptance?**

**Methods**

-Are the objectives of the study clearly articulated with a clear testable hypothesis stated?

-Is the study design appropriate to address the stated objectives?

-Is the population clearly described and appropriate for the hypothesis being tested?

-Is the sample size sufficient to ensure adequate power to address the hypothesis being tested?

-Were correct statistical analysis used to support conclusions?

-Are there concerns about ethical or regulatory requirements being met?

Reviewer #1: (No Response)

Reviewer #2: (No Response)

Reviewer #3: (No Response)

Reviewer #4: Yes to all except the last question. No ethical concerns and all regulatory requirements are met.

**Results**

-Does the analysis presented match the analysis plan?

-Are the results clearly and completely presented?

-Are the figures (Tables, Images) of sufficient quality for clarity?

Reviewer #1: (No Response)

Reviewer #2: (No Response)

Reviewer #3: (No Response)

Reviewer #4: Yes

**Conclusions**

-Are the conclusions supported by the data presented?

-Are the limitations of analysis clearly described?

-Do the authors discuss how these data can be helpful to advance our understanding of the topic under study?

-Is public health relevance addressed?

Reviewer #1: (No Response)

Reviewer #2: (No Response)

Reviewer #3: (No Response)

Reviewer #4: Yes

**Editorial and Data Presentation Modifications?**

Reviewer #1: (No Response)

Reviewer #2: (No Response)

Reviewer #3: (No Response)

Reviewer #4: Section 2.5 Study Risk of Bias Assessment: Entire paragraph is repeated.

Lines 295-296: “America (North America) displayed the lowest azithromycin resistance rate (2.4%), in contrast to Asia's highest rate of 32.2%. Similarly, North America exhibited the lowest clarithromycin resistance rate (0.7%), while Asia reported the highest rate at 30.8%. The Americas displayed the lowest resistance rate for erythromycin at 51.3%, whereas North America exhibited the highest rate at 94.1%”.In response to a comment during the first review, you mention that the data in Figure 5A for America’s represents data only from North America. So, why don’t the results for Americas (North America) in the text match the results in Figure 5A. For all other continents the data in the text matches the figures. Maybe I am missing something.

**Summary and General Comments**

Reviewer #1: (No Response)

Reviewer #2: (No Response)

Reviewer #3: (No Response)

Reviewer #4: Section 2.5 Study Risk of Bias Assessment: Entire paragraph is repeated.

Lines 295-296: “America (North America) displayed the lowest azithromycin resistance rate (2.4%), in contrast to Asia's highest rate of 32.2%. Similarly, North America exhibited the lowest clarithromycin resistance rate (0.7%), while Asia reported the highest rate at 30.8%. The Americas displayed the lowest resistance rate for erythromycin at 51.3%, whereas North America exhibited the highest rate at 94.1%”.In response to a comment during the first review, you mention that the data in Figure 5A for America’s represents data only from North America. So, why don’t the results for Americas (North America) in the text match the results in Figure 5A. For all other continents the data in the text matches the figures. Maybe I am missing something.

PLOS authors have the option to publish the peer review history of their article (what does this mean? ). If published, this will include your full peer review and any attached files.

**Do you want your identity to be public for this peer review?** For information about this choice, including consent withdrawal, please see our Privacy Policy .

Reviewer #1: No

Reviewer #2: No

Reviewer #3: No

Reviewer #4: No

---

## [Editor Report · Acceptance letter]

Dear dr Beig,

We are delighted to inform you that your manuscript, "Insights into Haemophilus Macrolide Resistance: A Comprehensive Systematic Review and Meta-Analysis," has been formally accepted for publication in PLOS Neglected Tropical Diseases.

Best regards,

Shaden Kamhawi

co-Editor-in-Chief

Paul Brindley

co-Editor-in-Chief
